# Urban and rural prevalence of tuberculosis in low- and middle-income countries: A systematic review and meta-analysis

Seyed Alireza Mortazavi[1,2], Nicole A. Swartwood[3], Nanki Singh[3], Melike Hazal Can[3], Hening Cui[3], Do Kyung Ryuk[3], Katherine C. Horton[4], Nicolas A. Menzies[3], Peter MacPherson[1]*

**1** School of Health & Wellbeing, University of Glasgow, Glasgow, United Kingdom, **2** Caldecot Centre, King's College Hospital NHS Foundation Trust, London, United Kingdom, **3** Department of Global Health and Population, Harvard T.H. Chan School of Public Health, Boston, Massachusetts, United States of America, **4** Department of Infectious Disease Epidemiology, London School of Hygiene and Tropical Medicine, London, United Kingdom

* Peter.MacPherson@glasgow.ac.uk

## Abstract

### Background

Urban and rural settings differ in key determinants of tuberculosis (TB) burden, including transmission dynamics, social and structural determinants, and healthcare access. However, understanding of urban and rural TB burden is limited, hindering implementation of public health interventions to end TB.

### Methods and findings

We conducted a systematic review and meta-analysis of urban and rural differences in adult pulmonary TB prevalence in low- and middle-income countries. We searched PubMed, Embase, Global Health, the Cochrane Library, Africa Index Medicus, LILACS, and SciELO for community-representative prevalence surveys conducted between 1st January 1993 and 14th October 2025. Studies focussing solely on symptomatic or healthcare-seeking individuals and those conducted in congregate settings like prisons, universities, and health facilities were excluded. Risk of bias was assessed using a tool for prevalence surveys. Bayesian multilevel meta-regression was used to estimate pooled urban-to-rural prevalence ratios (PR) for bacteriologically-confirmed and smear-positive TB overall, and by World Health Organization (WHO) region. We also investigated time trends in the urban-to-rural prevalence ratio, and associations between urban-to-rural prevalence ratios and survey-level risk of bias (not low versus low), TB screening algorithm (whether used symptom screening for sputum eligibility), national TB incidence, percentage of population living in urban areas, and representativeness of prevalence surveys (national versus sub-national). To estimate the number of people with prevalent TB

**Data availability statement:** The extracted datasets and code used in the analysis are available from Github [https://github.com/peter-macp/urban-to-rural/releases/tag/v1.0.0] and archived in Zenodo [https://doi.org/10.5281/zenodo.19145846].

**Funding:** PM was funded by Wellcome (304666/Z/23/Z) and an NIHR Global Health Research Professorship (NIHR304311). KCH and PM are supported by the UK FCDO (Leaving no-one behind: transforming gendered pathways to health for TB). KCH is supported by the U.S. National Institutes of Health (R-202309-71190). This research has been partially funded by UK aid from the UK government (to KCH and PM). The funders (Wellcome, NIHR, UK FCDO, and U.S. National Institutes of Health) had no role in the study design, data collection and analysis, decision to publish, or preparation of the manuscript.

**Competing interests:** I have read the journal's policy and the authors of this manuscript have the following competing interests: PM is an Academic Editor on PLOS Medicine's editorial board. All other authors declare that no competing interests exists.

**Abbreviations:** CrI, credible interval; LMICs, low- and middle-income countries; PR, prevalence ratios; PRISMA, Preferred Reporting Items for Systematic Reviews and Meta-Analyses; TB, tuberculosis; WHO, World Health Organization.

in urban and rural areas in study countries, and how these have changed between 2000 and 2024, we fitted a Bayesian multivariate model to WHO incidence and case detection ratio data and combined these estimates with assumptions about the duration of treated and untreated TB and the distribution of urban and rural populations.

We included 47 surveys conducted between 2000 and 2024, encompassing 2,454,443 participants. The pooled urban-to-rural PR of bacteriologically-confirmed TB was 1.09 (95% credible interval [CrI]: 0.90, 1.30) and was 1.24 (95% CrI: 0.94, 1.61) for smear-positive TB. However, there were substantial differences between WHO regions: averaged across the 24 year study period the African Region had higher urban bacteriologically-confirmed prevalence (PR 1.18, 95% CrI: 0.91, 1.52), while the Western Pacific Region (PR 0.85, 95% CrI: 0.64, 1.07) and South-East Asia Region (PR 0.86, 95% CrI: 0.67, 1.08) had broadly similar urban and rural prevalence. Time trends indicated an increase in the overall bacteriologically-confirmed urban-to-rural prevalence ratio between 2000 and 2024, with a mean PR increase of 2.4% (95% CrI: −0.8%, 6.0%) per year. We estimated that, for 2024 in the 26 represented study countries (combined population: 2.24 billion [48.3%] urban; 2.40 billion [51.7%] rural), 49% (6.6 million, 95% CrI: 4.2, 12.0 million) of prevalent TB was in urban areas, and 51% (6.8 million, 95% CrI: 4.2, 12.0 million) in rural areas. Within countries, there were striking changes in the urban and rural distribution between 2000 and 2024, with the share of urban cases increasing in nearly all countries. The main limitations include lack of unified definitions for urban and rural areas, and absence of data for some global regions (e.g., Americas and Europe).

## Conclusion

Between 2000 and 2024, TB epidemics have become increasingly urbanised, both in proportional and absolute terms, although with considerable variation in timing across countries and regions. Public health approaches tailored to urban and rural TB epidemiology and demography will be required to end TB.

---

Author summary

### Why was the study done?

- Tuberculosis (TB) risk and access to care differ substantially between urban and rural settings, but the global magnitude, direction, and evolution of these differences have not been systematically quantified.

- Rapid urbanisation in low- and middle-income countries is reshaping population structure and disease risk, yet its implications for where people with TB live—and how TB control should be targeted—remain unclear.

### What did the researchers do and find?

- We systematically reviewed 47 national and sub-national TB prevalence surveys from 26 low- and middle-income countries (2000–2024) and used Bayesian multilevel meta-regression to estimate urban-to-rural TB prevalence ratios overall and by World Health Organization region.

- Overall TB prevalence was similar in urban and rural areas, but patterns varied markedly by region: averaging across the 24 year study period, prevalence was higher in urban areas in Africa, and broadly similar in the Western Pacific and in South-East Asia; despite this heterogeneity in timing towards urban predominance, the absolute and proportional share of TB occurring in urban populations increased in nearly all countries over time.

### What do these findings mean?

- TB epidemics are becoming increasingly urbanised in absolute and proportional terms, driven by demographic change and region-specific epidemiological factors rather than urbanisation alone.

- Ending TB will require context-specific strategies that address transmission and social determinants in cities—particularly in rapidly growing urban populations—while also strengthening access to diagnosis and care in rural and ageing communities.

- Limitations include the absence of data from some WHO regions, especially the Americas and Europe, variation in definitions of urban and rural areas between studies, and limited prevalence survey data from the post-COVID-19 period.

## Introduction

Tuberculosis (TB) remains a leading infectious cause of death worldwide, with incidence and mortality concentrated in low- and middle-income countries (LMICs) [1,2]. In 2014, the World Health Organization (WHO) set an ambitious goal to end TB by 2035, yet substantial progress is needed to meet this goal [1].

Understanding variation in TB burden both between and within countries and regions is essential for developing effective public health strategies to address TB determinants and accelerate TB care and prevention [3]. Urban and rural environments differ substantially in the factors that shape TB transmission, healthcare access, and health-seeking behaviours [4]. Urban areas may experience higher TB prevalence due to greater population density, higher rates of HIV infection, and increased exposure to crowded congregate settings and poor outdoor air-quality [5]. In contrast, people living in rural areas can face barriers such as limited healthcare access, delayed diagnosis, and treatment challenges, as well as higher rates of indoor air pollution [6,7]. For instance, in India, whilst 72% of the population lives in rural areas, only 40% of healthcare workers are stationed there [8]. TB epidemiology may also be affected by high rates of recent rural-to-urban migration [9], with rural populations experiencing more rapid ageing [10].

Although some national TB prevalence surveys have found differences in TB prevalence between urban and rural areas (e.g., [11]), the magnitude of this difference has not been systematically quantified, and there is little understanding of country and regional patterns and determinants.

Therefore, we conducted a systematic review and meta-analysis using data from national and sub-national TB prevalence surveys conducted in LMICs. We compared urban and rural TB prevalence and estimated burden and secular trends of urban and rural TB within LMICs and WHO regions to inform targeted resource allocation and public health approaches to TB elimination.

## Methods

We conducted a systematic review and meta-analysis of differences in TB prevalence between urban and rural areas. This study is reported as per the Preferred Reporting Items for Systematic Reviews and Meta-Analyses (PRISMA) guideline (S1 Checklist) [12]. Our protocol was prospectively registered (PROSPERO number: CRD42024503853) [13].

### Inclusion and exclusion criteria

We included studies published between 1st January 1993 (the year when WHO declared TB to be a public health emergency of international concern) and 14th October 2025 that reported the prevalence of pulmonary TB in both rural and urban adult populations (15 years or older, or where the majority of participants where 15 years or older and it was not possible to disaggregate prevalence estimates for people younger than 15 years) in LMICs (as defined by the World Bank country groups "lower income", "lower-middle income", and "upper-middle income" [14]). Studies published in English, French, Portuguese, and Spanish were included. We included pre-intervention surveys conducted as part of community cluster randomised trials but excluded post-intervention surveys. Eligible studies reported prevalence estimates—defined as numerator and denominator counts, or unadjusted or adjusted prevalence rates—stratified by urban and rural populations, or provided sufficient data for their calculation. We excluded studies that reported only extra-pulmonary TB or TB prevalence exclusively among children under the age of 15 years. Studies that included data only for symptomatic or healthcare-seeking individuals, such as studies of case notifications, were excluded. We also excluded studies conducted in congregate settings, including prisons, universities, and health facilities. Full inclusion and exclusion eligibility criteria are provided in S1 Table.

### Search strategy

Studies were identified by searching PubMed, Embase, Global Health, the Cochrane Library, Africa Index Medicus, LILACS, and SciELO. Search terms were defined to identify all relevant studies by utilising a combination of MeSH terms and keywords related to 1) tuberculosis, 2) prevalence, and 3) LMICs (S2 Table). We additionally considered all studies from a previous systematic review of sex differences in TB prevalence [15], WHO reports of prevalence surveys sourced from the Global TB Report, as well as the abstracts of the 2023 Union World Conference on Lung Health.

 This systematic review was managed using Covidence [16]. After removal of duplicates, titles and abstracts were independently reviewed for relevance by pairs of reviewers, allocated randomly from a pool of eight team members with resolution of discrepancies by a third reviewer or team discussion. Studies published in French, Portuguese, and Spanish were translated using DeepL. Studies included for full-text review were independently assessed against inclusion and exclusion criteria by two reviewers, randomly allocated in pairs from a team of eight, with recording of reasons for exclusion. Where there were multiple publications relating to one prevalence survey, we grouped these, reviewed all together, and extracted data from the combined set. Extraction was done in duplicate by pairs of reviewers using a pre-tested form. Data extracted included: title and identification information; survey methodology; and screening methods; case definitions; and prevalence survey results. A consensus reviewer, allocated at random, resolved discrepancies in data extraction, with further group discussion, if required.

### Definitions

We defined bacteriologically-confirmed TB in a study participant as at least one positive sputum result for *Mycobacterium tuberculosis* on testing by smear microscopy, culture, or molecular testing (e.g., Xpert MTB/RIF). Smear-positive TB was defined by a positive smear microscopy result, regardless of other results. Where study definitions varied (for example, requiring more than one positive test to classify a case as bacteriologically-confirmed TB), we adhered to the definitions

used in each study. We used study definitions of "urban" and "rural". If a study reported more than two urbanicity categories, but these categories could be grouped into urban and rural, they were aggregated accordingly (e.g., "state-urban" and "regional-urban" were grouped as urban). If a study reported an urbanicity category that could not be clearly classified as urban or rural, data for this category were excluded.

## Outcomes

The primary outcome was the urban-to-rural TB prevalence ratio, defined as the ratio of bacteriologically-confirmed TB prevalence in urban versus rural areas. In secondary analysis we estimated variation in ratios stratified by WHO global region and survey country, and the urban-to-rural prevalence ratio of smear-positive TB. We also conducted meta-regression to investigate predictors of urban-to-rural prevalence ratio (overall, and by WHO region). Predictors included the proportion of the study country population who resided in urban areas [17] and country-level TB incidence in the prevalence survey year [1]. Finally, we constructed epidemiological models to estimate the numbers of people with prevalent TB in urban and rural areas, and how these have changed over time.

## Risk of bias assessment

Risk of bias was assessed using a tool developed by Hoy *and colleagues* for prevalence surveys [18]. Two reviewers each independently classified studies as low, medium, or high risk of bias across domain questions and made an overall risk of bias assessment (high, moderate, low, and unknown), with discrepancies resolved by a third consensus reviewer.

## Statistical analysis

We summarised study characteristics, including setting, survey methodologies, TB screening and diagnosis methods, and case definitions. In initial descriptive analysis, we extracted or calculated (where not reported) the crude urban and rural prevalence for each study, based on the reported number of prevalent TB cases and population denominators, or rates, and calculated the crude urban-to-rural prevalence ratio. Because prevalence surveys are usually conducted within clusters, the reported 95% confidence intervals are often adjusted for clustering, typically using robust standard errors or regression models [19]. Where available, we extracted these cluster-adjusted confidence intervals. Moreover, typically in prevalence surveys certain population groups, usually men and younger people, are under-represented and surveys often undertake weighted analysis to adjust prevalence estimates for this under-representation [19]. Where reported, we additionally extracted adjusted urban and rural TB prevalence estimates from a WHO-recommended regression approach, which uses multiple imputation and inverse probability weighting to address sampling bias [19]. To pool estimates of the urban-to-rural prevalence ratios, we preferentially used survey point estimates and 95% confidence intervals based on adjusted/imputed prevalence rates, followed by crude prevalence rates, and using counts of numerators and denominators where neither prevalence rate was available (data hierarchy shown in S1 Fig).

We constructed hierarchical Bayesian meta-regression models [20] to estimate the pooled log-odds of the urban-to-rural prevalence ratio for bacteriologically-confirmed and smear-positive pulmonary TB, weighted for precision by the log of the survey-specific prevalence ratio standard error, and with a hierarchical random-effects grouping term for each prevalence survey, nested within countries where there was more than one survey per country (S1 Text). Subsequent meta-regression models included terms for WHO region (with countries from WHO regions with few studies excluded from regional pooling), estimated country TB incidence in the prevalence survey year [1], survey year (or mid-year where spanned one than 1 year), the percentage of country population estimated to be living in urban areas [17], whether symptom screening was used to identify participants for sputum testing, and risk of bias. Priors were weakly informative. We retained 4,000 posterior draws for each parameter from the posterior and exponentiated and summarised these results to obtain mean posterior prevalence ratios and 95% credible intervals (CrI).

PLOS Medicine

To estimate the number and percentage of people with prevalent TB in urban and rural areas in study countries, and how these have changed between 2000 and 2024, we fitted a Bayesian multivariate model in which WHO TB incidence and case detection ratio were modelled jointly as correlated outcomes. The model included random-effects for surveys nested within country, allowing country-specific and time-varying estimates, including for countries with sparse regional representation. We combined these estimates with assumptions about the duration of treated and untreated TB (stratified by smear status) [21] and the distribution of urban and rural populations (S1 Text) [17].

Models were fit using the brms interface (version 2.22.0) [22] to Stan (version 2.37.0) [23]. We assessed model convergence using $\hat{R}$ statistics, effective sample size measures, trace plots of chains, and posterior predictive plots. Model predictions and summaries used the tidyverse [24] and tidybayes [25]. All analyses were conducted using R version 4.5.0 [26]. The extracted datasets and code used in the analysis are available from Github [https://github.com/petermacp/urban-to-rural/releases/tag/v1.0.0] and archived in Zenodo [https://doi.org/10.5281/zenodo.19145846].

## Results

Our search identified 11,558 records, of which 10,211 were screened after duplicate removal. Following abstract screening, 218 studies underwent full-text review, and 46 studies reported TB prevalence disaggregated by urban and rural setting and were included in this analysis [11,27–71]. One study in China reported two separate prevalence surveys [54], giving a total of 47 prevalence surveys that contributed to quantitative analysis (Fig 1).

### Study characteristics

The 47 included prevalence surveys were conducted between 2000 and 2024 across 27 countries from four WHO regions [11,27–71]. The African (n = 18) [30,31,33–35,39,41,48–50,55,56,58–60,64,65,70] and South-East Asia Regions (n = 15) [28,36,38,43,45–47,52,53,57,61–63,68,71] contributed the largest number of surveys, followed by the Western Pacific Region (n = 11) [11,27,29,32,40,42,44,51,54,66,67] and the Eastern Mediterranean Region (n = 2) [37,69]. India (n = 7) [28,46,47,52,53,57,62], China (n = 5) [11,32,42,54], and Ethiopia (n = 4) [39,59,64,65] had the greatest number of surveys among individual countries. Overall, 32/47 (68.1%) of surveys were nationally representative surveys, most of which used multistage cluster sampling. Details of included surveys are provided in S3 Table.

In 36/47 (76.6%) surveys, participants were eligible for sputum testing if they reported TB symptoms or had abnormalities on chest radiography, eight (17.0%) screened with symptom screening alone, one (2.1%) conducted symptom screen followed by chest X-ray if positive, and the remaining two (4.3%) collected sputum samples from all participants without a screening stage.

Across the 47 surveys 2,508,921 individuals were eligible for screening (S4 Table). Screening eligibility was not always reported stratified by urban or rural status, but where reported, 39.8% of screening-eligible participants were from urban settings, and 60.2% were from rural settings. Overall, 97.8% of eligible individuals participated in screening, with 39.0% from urban settings and 61.0% from rural settings, where reported. Bacteriologically-confirmed TB was diagnosed in 7,509 participants (where reported: urban: 39.9%; rural: 60.1%). Overall, 3,185 had smear-positive TB (where reported: urban: 35.7%; rural: 64.3%) and 3,803 had culture-positive TB (where reported: urban: 35.7%; rural: 64.3%).

### Urban-to-rural prevalence of bacteriologically-confirmed TB

Across all 47 surveys, reported crude urban-to-rural prevalence ratios ranged from 0.20 (China, Shandong Province, 2010) to 2.71 (Malawi National TB Prevalence Survey, 2013) (Fig 2). The lowest crude ratios—below 0.5—were reported in eight surveys, all conducted in China and India. In contrast, five of the six surveys reporting urban-to-rural prevalence ratios greater than two were from WHO African Region, with the other being Sudan from the Eastern Mediterranean Region. Overall, 25/47 (53.2%) of surveys reported a crude urban-to-rural ratio greater than one.

PLOS Medicine

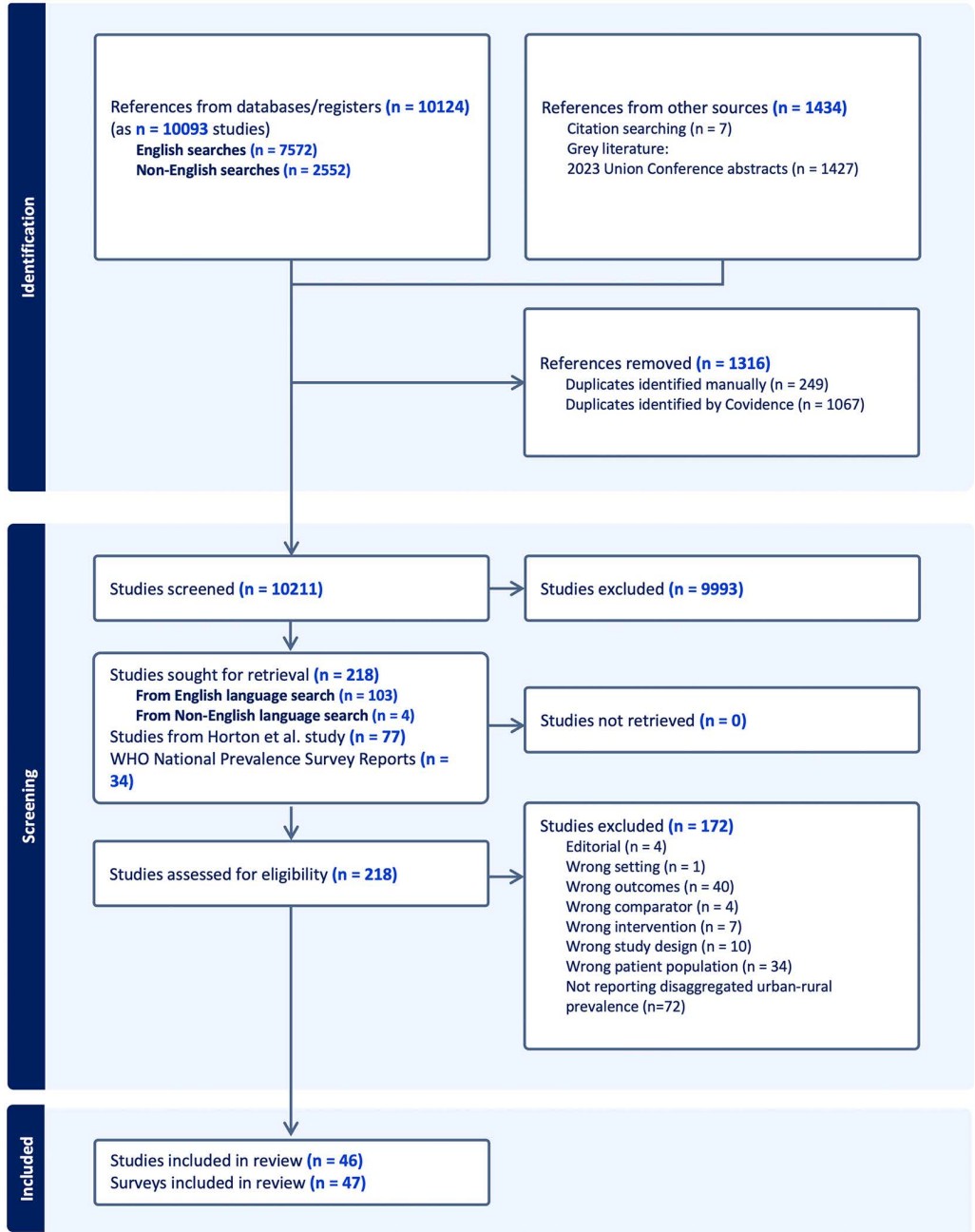

**Fig 1. PRISMA Flow Chart.** Horton and colleagues refers to Horton KC, MacPherson P, Houben RM, White RG, Corbett EL. Sex differences in tuberculosis burden and notifications in low- and middle-income countries: a systematic review and meta-analysis. PLos Med. 2016 Sep;13(9):e1002119. https://doi.org/10.1371/journal.pmed.1002119. PMID: 27598345; PMCID: PMC5012571.

In our main model, which excluded two studies from the Eastern Mediterranean Region with disparate crude estimates, the pooled urban-to-rural prevalence ratio for bacteriologically-confirmed TB averaged across the study time period was 1.09 (95% CrI: 0.90, 1.30) (Fig 2). This is equivalent to prevalence rates being 9% higher in urban settings, but with an uncertainty interval including 0%. The magnitude and direction of the urban–rural difference varied by WHO region.

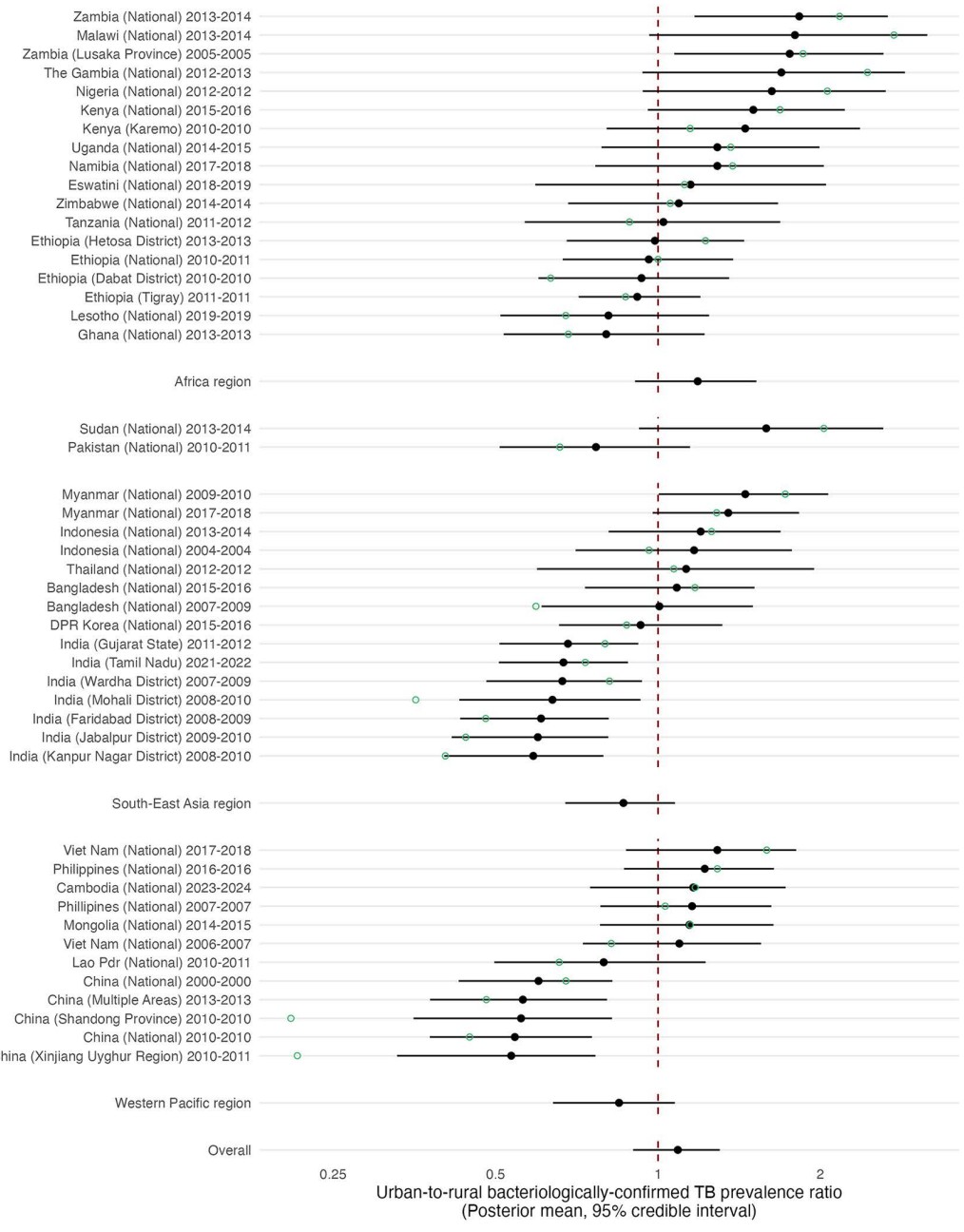

**Fig 2. Forest plot of urban-to-rural prevalence ratios of bacteriologically-confirmed tuberculosis.** Green points are crude survey urban-to-rural prevalence ratios, and black points and error bars posterior means and 95% credible intervals from multilevel meta-analysis models. Pooled estimates were not estimated for Eastern Mediterranean Region as these were not included in multilevel meta-analysis model. Overall pooled estimate includes data from African Region, South-East Asia Region, and Western Pacific Region, but not Eastern Mediterranean Region.

Averaged across the study period, the African Region showed higher prevalence in urban settings (PR 1.18, 95% CrI: 0.90, 1.52), equivalent to 18% higher prevalence in urban areas), whereas in the Western Pacific Region (PR 0.85, 95% CrI: 0.64, 1.07) and South-East Asia Region (PR 0.86, 95% CrI: 0.67, 1.08) prevalence was broadly similar. Comparing between WHO regions, the average urban–rural prevalence ratio was higher in the African Region than in the Western

Pacific Region (ratio of prevalence ratios 1.43, 95% CrI: 1.00, 2.06) and higher than in the South-East Asia Region (1.39, 95% CrI: 0.99, 1.98), while differences between South-East Asia and the Western Pacific were less certain (1.03, 95% CrI: 0.74, 1.43).

## Change in the urban-to-rural prevalence ratio over time

Time trend analyses were consistent with an increase in the overall urban-to-rural prevalence ratio of bacteriologically-confirmed TB between 2000 and 2024, with a mean increase of 2.4% (95% CrI: −0.8%, 6.0%) per year, but with substantial uncertainty (Fig 3). By 2024, there was an 96.6% probability that the overall urban-to-rural prevalence ratio exceeded one. Estimates of the average annual change in the PR within individual WHO regions were imprecise: African Region 1.7%, −5.4%, 8.3%); South-East Asia Region (3.6%, −1.3%, 9.4%); Western Pacific (4.1%, −0.4%, 9.7%). By 2024, the posterior probability that the urban-to-rural prevalence ratio exceeded one was 82.3% in the African Region, 80.6% in the South-East Asia Region, and 83.8% in the Western Pacific Region.

In univariate analysis, among the 45 surveys in 25 LMICs in the African, South-East Asia, and Western Pacific Regions, there was evidence that nationally representative prevalence surveys had higher urban-to-rural prevalence ratios than sub-national surveys (multiplicative effect: 1.84, 95% CrI: 1.13, 3.02) (Table 1). The magnitude and directionality of associations were consistent across all three global regions. Associations estimated for other predictors were consistently weak.

## Urban-to-rural prevalence of smear-positive TB

Thirty-one surveys with 1,433,666 total participants (36.7% urban) reported urban–rural smear-positive TB prevalence. The pooled urban-to-rural smear-positive TB prevalence was 1.24 (95% CrI: 0.94, 1.61). The estimated smear-positive prevalence ratio was 1.43 (95% CrI: 1.03, 1.96) in WHO African Region, 1.05 (95% CrI: 0.71, 1.47) in South-East Asia Region, and 1.04 (95% CrI: 0.60, 1.59) in the Western Pacific Region. Overall, time trends were more pronounced than for bacteriologically-confirmed TB; there was an estimated 15.4% (95% CrI: 3.5%, 30.7.2%) annual increase in the

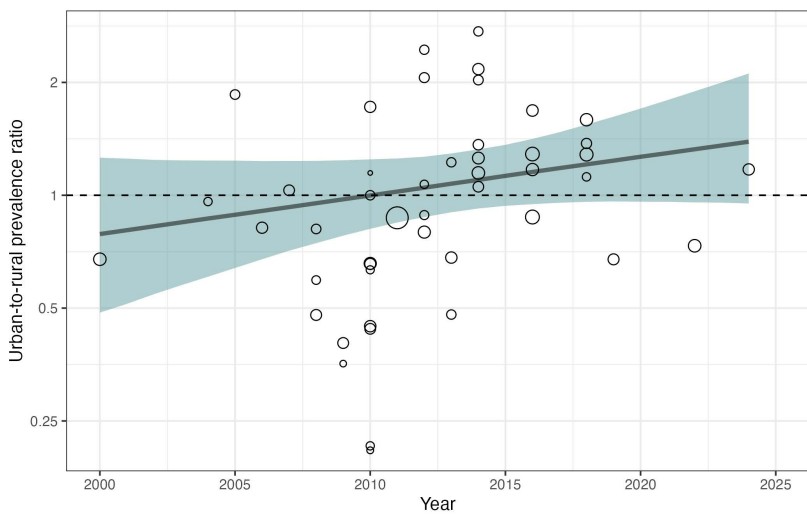

**Fig 3. Time trend in the urban-to-rural prevalence ratio for bacteriologically-confirmed pulmonary tuberculosis: 2000-2024.** Points are crude prevalence survey estimates at survey midpoint.

**Table 1. Multiplicative effect of covariates on urban-to-rural prevalence ratio of bacteriologically-confirmed TB prevalence.**

| | Global (25 LMICs) | | African Region | | South-East Asia Region | | Western Pacific Region | |
|---|---|---|---|---|---|---|---|---|
| | Univariate | Multivariable | Univariate | Multivariable | Univariate | Multivariable | Univariate | Multivariable |
| Risk of bias (not low vs. low) | 1.06 (0.61–1.77) | 1.02 (0.57–1.90) | 1.03 (0.63–1.58) | 1.03 (0.62–1.71) | 1.10 (0.68–1.76) | 0.97 (0.54–1.64) | 1.03 (0.58–1.66) | 1.05 (0.60–1.95) |
| National TB incidence† (per 100/100,000 increase) | 1.15 (0.93–1.44) | 1.09 (0.86–1.37) | 1.10 (0.92–1.29) | 1.07 (0.90–1.25) | 1.16 (0.96–1.43) | 1.09 (0.89–1.35) | 1.17 (1.02–1.36) | 1.11 (0.95–1.31) |
| Nationally representative prevalence survey (Yes vs. No) | 1.84 (1.13–3.02) | 1.80 (1.04–3.25) | 1.63 (1.04–2.35) | 1.73 (1.01–2.81) | 1.84 (1.30–2.59) | 1.85 (1.17–3.05) | 1.98 (1.23–3.38) | 1.82 (1.10–3.24) |
| Percent of population in urban areas† (per % point increase) | 1.00 (0.97–1.03) | 1.00 (0.97–1.03) | 1.01 (0.99–1.02) | 1.00 (0.98–1.02) | 1.00 (0.98–1.02) | 1.00 (0.97–1.02) | 1.00 (0.98–1.01) | 0.99 (0.97–1.01) |
| Symptom screening used for sputum eligibility (Yes vs. No)¶ | 0.96 (0.35–2.17) | 1.03 (0.32–3.25) | 1.00 (0.40–2.13) | 1.06 (0.35–3.22) | 0.96 (0.38–2.13) | 1.07 (0.36–3.21) | 0.93 (0.32–2.17) | 0.97 (0.26–3.22) |

Eastern Mediterranean Region countries not included. Multivariable models adjusted for all covariates. LMICs: low- and middle-income countries.

†In the year (or mid-year) in which the prevalence survey was conducted. ¶Compared to surveys where chest X-ray was used for screening.

urban-to-rural prevalence ratio of smear-positive TB between 2000 and 2024, with a 99.7% posterior probability that the prevalence ratio was greater than one in 2024.

### Burden of bacteriologically-confirmed TB in urban and rural populations

Between 2000 and 2024, urban and rural populations increased substantially in the 26 countries represented in the review (S2 Fig). In some countries, especially China, Indonesia, and Thailand, there was rapid urbanisation (S3 Fig). By 2024, we estimated that there were 4.63 billion people in these 26 study countries, with 2.24 billion (48.3%) in urban areas, and 2.40 billion (51.7%) in rural areas.

Multivariate modelled estimates of WHO TB incidence and case detection ratio data between 2000 and 2024 are shown in S4 and S5 Figs, and country-specific estimates of the urban-to-rural prevalence ratio of bacteriologically-confirmed TB in S6 Fig. All countries had declining TB incidence, increasing case detection (excluding in a small number of countries during the COVID-19 pandemic in 2020–2021), and an increasing urban-to-rural prevalence ratio over time.

By modelling changes in TB incidence and case detection data, and incorporating country-specific population structure and urban-to-rural prevalence ratio estimates, we estimated that in these 26 countries in 2024 there were 13.4 million (95% CrI: 8.5, 22.7 million) people with prevalent TB. Of these, 49% (6.6 million, 95% CrI: 4.0, 11.3 million) were in urban areas, and 51% (6.8 million, 95% CrI: 4.2, 12.0 million) were in rural areas. In absolute terms, in 2024, India, Pakistan, and Indonesia had the greatest numbers of both urban and rural people with prevalent TB in 2024 (S5 Table). Within WHO African Region, the greatest number of people with prevalent TB in rural and urban areas were in Nigeria, Ethiopia, and Tanzania.

Between 2000 and 2024, there were striking changes in the distribution of TB between urban and rural populations within countries, both in percentage terms (Fig 4), and by absolute numbers (S7 Fig). In all countries, the percentage of people with prevalent TB in rural areas declined, and in urban areas increased. By 2024, the median percentage of prevalent cases was higher in urban than in rural areas in 12 of the 26 countries (Fig 4).

### Discussion

In this systematic review and meta-analysis of 46 national and sub-national TB prevalence surveys from 26 low- and -income countries (together representing more than 4.6 billion people in 2024), we found that TB epidemics are becoming increasingly urbanised in both proportional and absolute terms, driven by rapidly urbanising demographic trends. Meta-analysis of smear-positive TB prevalence showed a faster rate of urbanisation than for bacteriologically-confirmed

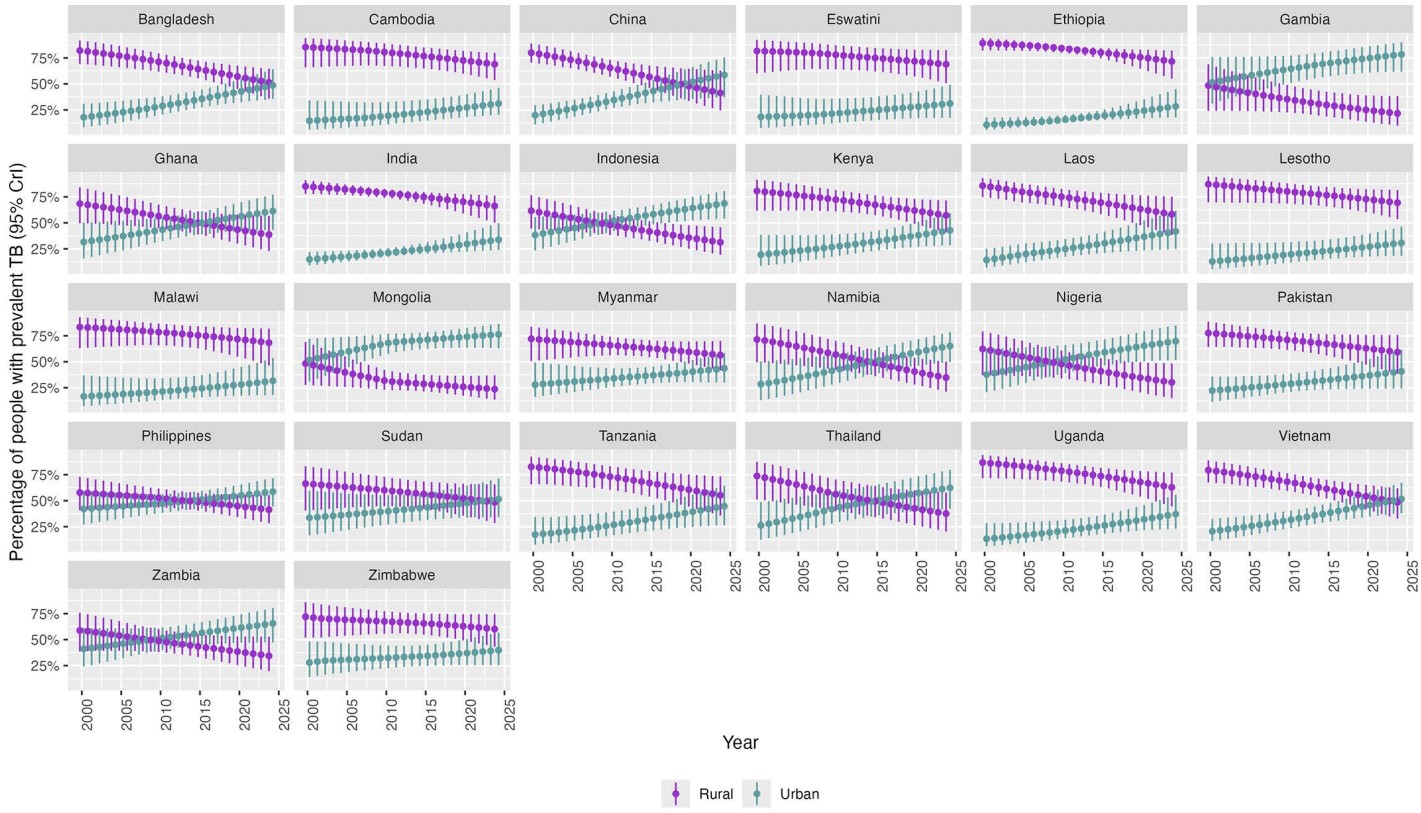

**Fig 4. Percentage of people with prevalent TB in urban and rural areas in 26 countries: 2000-2024.** Estimated from a Bayesian multivariate regression model, which jointly estimated TB incidence and case detection ratio, and combined with assumptions around duration of infectiousness, distribution of urban and rural populations, with modelled estimates of urban and rural TB prevalence by year.

TB, emphasising the need to accelerate efforts to interrupt TB transmission within cities, as well as adapt to likely changing epidemiology in rural areas. These findings highlight the importance of understanding local TB epidemiology to tailor public health strategies to context, which will be essential for accelerating progress toward the End TB Strategy targets and ensuring equitable health gains across all populations.

Urbanisation in low- and middle-income countries has been one of the major demographic shifts of the past century, with profound implications for the transmission and control of communicable diseases such as TB [72]. Between 2000 and 2024, the total populations of all study countries increased rapidly, with the proportion living in urban areas rising substantially in 25 of 26 countries, Zimbabwe being the exception. For example, Uganda's urban share grew by 85.2% (a 12.6 percentage-point increase, from 14.8% to 27.4%) and China's by 82.7% (a 29.7 percentage-point increase from 35.9% to 65.5%). These shifts are largely driven by rural-to-urban migration, and in many high TB incidence countries this has resulted in substantial fractions of the urban population living in unplanned settlements [72]. Rapid urban population growth, when combined with overcrowding, substandard housing, and under-resourced health systems, creates environments conducive to the transmission of infectious diseases, including TB [5,73].

Coinciding with these rapid demographic changes, we found that the urban-to-rural prevalence ratio of bacteriologically-confirmed TB likely increased over the 24 years during which surveys were conducted, with the rise being particularly pronounced for smear-positive disease. This trend cannot be explained by population redistribution alone: the ratio reflects a relative increase in prevalence, not simply a growing concentration of people in cities. In other words, TB burden within

urban populations increased relative to rural populations, over and above demographic shifts toward urban living. However, this overall pattern masked substantial regional heterogeneity, with the direction and magnitude of the urban–rural gradient differing across Africa, South-East Asia, and the Western Pacific. The timing of transitions to predominantly urban TB epidemics also varied considerably between countries, reflecting differences in demographic change, health system capacity, and social conditions [73]. These findings suggest that demographic and epidemiological processes are acting in concert, with urban growth amplifying—but not solely determining, as emphasised by the absence of association with urban population fraction in the meta-regression—the increasing concentration of TB in cities.

Regional patterns revealed substantial heterogeneity in timing of the shift of relative burden of TB from rural to urban populations. In the African Region, urbanisation of TB epidemics has occurred earlier, with a 83% posterior probability the urban-to-rural prevalence ratio exceeded one in 2024. In contrast, in the Western Pacific Region, and South-East Asia regions overall urban and rural prevalence were broadly similar averaged over the study period, although by 2024 rapid urbanisation of TB epidemics was also occurring in these regions. Despite higher rural prevalence in many Western Pacific countries, rapid urban population growth—particularly in China, Indonesia, and the Philippines—means that the absolute number and proportion of people with TB living in cities has become larger than in rural areas, and continues to rise. These shifts are compounded by rapid population ageing in the region [4,10], which disproportionately affects rural areas, as younger people migrate to cities for employment.

In countries in the African Region, the higher prevalence of TB in urban compared with rural populations is likely to reflect the combined influence of overlapping epidemics and demographic changes. Cities have been the epicentres of HIV epidemics in Africa [74,75], and HIV remains the strongest risk factor for progression from TB infection to disease [76]. The co-location of dense urban populations and concentrated HIV epidemics—along with other socioeconomic TB determinants—has therefore amplified TB transmission and disease incidence in cities. At the same time, many African countries are experiencing rapid population growth and urbanisation, with the most substantial increases occurring among younger age groups [77]. This demographic expansion fuels the absolute number of individuals at risk of TB, while high levels of mobility between rural and urban areas further complicate transmission dynamics and case detection [5].

In the South-East Asia Region, urban and rural TB prevalence appeared broadly similar, but this masks the rapid changes over time and enormous influence of India, which accounts for more than a quarter of the global TB burden [1]. India's rapid urbanisation has produced vast, densely populated cities where TB transmission is sustained by poverty, overcrowding, and air pollution, yet large rural populations remain at risk due to limited access to timely diagnosis and care. Other countries in the region, including Indonesia, Myanmar, and Bangladesh, likely show similar dual challenges of intense transmission in urban centres and persistent barriers to care in rural areas. The absence of a strong average urban–rural gradient in regional analyses for South-East Asia and the Western Pacific, therefore, reflects the later timing of transition to urban epidemics compared to in Africa, the scale of India's contribution, and the coexistence of high TB risks in different settings. For TB prevention and care in South-East Asia, addressing the needs of urban poor populations—particularly in informal settlements—while also strengthening rural primary care and social protection systems will be critical to reducing the overall burden.

Taken together, these dynamics suggest that TB control strategies must be informed by both demographic change and epidemiological context. In urban areas, interventions should prioritise the social determinants of TB—such as overcrowding, substandard housing, and poor air-quality [4]—while strengthening case-finding approaches, including targeted screening in high-risk groups [78], alongside systematic contact investigation and preventive therapy [79]. In high HIV-prevalence countries, particularly in Africa, public health efforts to combat TB in cities must integrate closely with HIV and other services [80] and adapt to the realities of youthful, highly mobile urban populations, with interventions spanning early diagnosis, preventive therapy, and targeted social support. In rural areas, where TB exposure may be more distal [81] and populations are ageing, strategies that improve access to routine screening within healthcare services, particularly for older adults, and that address risk factors for disease progression such as undernutrition [82], will likely be needed.

 

This study has several limitations. There is no unified global definition of urban and rural areas, and countries and regions often use varying criteria [73]. To estimate urban and rural TB burden, we used population estimates from the United Nations and World Bank; these may misestimate population growth, and particular rates of urbanisation. Only two studies were available from the Eastern Mediterranean Region (Pakistan and Sudan), each providing markedly different estimates of the urban-to-rural TB prevalence ratio; additional data from this, and other under-represented WHO regions with high TB incidence countries (e.g., Region of the Americas, European), are needed. To estimate numbers of people with prevalent TB and temporal trends, we relied on WHO incidence and case detection data combined with assumptions about disease duration recognising that for Nigeria and Bangladesh, WHO estimates of incidence are uncertain. Although this approach has been widely used, including by WHO, transmission modelling methods may provide more robust estimates. Approaches used by WHO to estimate TB incidence and case detection make use of prevalence survey data in some settings, which could create dependences between data sources if we also used these survey data for our analysis. For this reason, we only used prevalence survey data to estimate urban–rural contrasts, and relied on WHO data to estimate overall prevalence levels; together with Bayesian partial pooling across countries, we believe this is unlikely to materially bias sub-national TB burden estimates. We additionally projected estimates for numbers of people with prevalent TB beyond 2019–2024. All but one included prevalence surveys were conducted before 2019; further estimates from the post-COVID-19 era are not available, and future prevalence surveys may be constrained by reductions in international funding [83]. Finally, although our primary analyses used WHO-recommended survey estimates that adjust for differential participation through multiple imputation and inverse probability weighting (where available), these methods assume that nonparticipation is independent of prevalent tuberculosis conditional on observed covariates; if participation is additionally related to TB risk—for example, differently across urban and rural settings—some residual bias in prevalence estimates may remain.

In summary, TB epidemics are increasingly concentrated in urban populations both in proportional and absolute terms, likely driven by highly dynamic population changes over the past 24 years. In Africa, earlier high urban TB prevalence, and proportions and numbers of people with prevalent TB in urban areas are likely to have been amplified by rapid urbanisation, overlapping HIV epidemics, and rapid growth of young populations. In South-East Asia, and the Western Pacific the later shift to higher urban-to-rural prevalence ratio is moderated by the dominant contribution of the large rural population in India, China, Indonesia, and the Philippines and where both urban transmission and rural barriers to care shape the epidemic; yet rapid urban population growth and ageing of the rural population, has rapidly increased the absolute number of urban people with prevalent TB. These findings highlight that both demographic change and local epidemiological context drive TB burden, underscoring the need for TB care and prevention strategies tailored to the specific social, demographic, and health system contexts of each setting.

## Supporting information

**S1 Checklist. PRISMA Checklist.** PRISMA checklist. Page MJ, McKenzie JE, Bossuyt PM, Boutron I, Hoffmann TC, Mulrow CD, and colleagues. The PRISMA 2020 statement: an updated guideline for reporting systematic reviews. BMJ 2021;372:n71. https://doi.org/10.1136/bmj.n71. This work is licensed under CC BY 4.0. To view a copy of this license, visit https://creativecommons.org/licenses/by/4.0/.
(PDF)

**S1 Table. Inclusion and exclusion eligibility criteria.** Summary of inclusion and exclusion eligibility criteria.
(DOCX)

**S2 Table. Search Strategy.** Database search strategy.
(DOCX)

**S1 Fig. Hierarchy of TB prevalence survey estimates used in meta-analysis.** Summary of hierarchical approach to use of study prevalence estimates for inclusion in meta-analysis.
(TIFF)

**S1 Text. Supplemental methods for estimating urban-to-rural prevalence ratio, and prevalence and burden of TB in urban and rural populations.** Description of additional statistical methods.
(DOCX)

**S3 Table. Summary characteristics of included studies.** Summary of studies included in systematic review.
(CSV)

**S4 Table. Descriptive Summary of Included Studies.** Summary of TB screening and prevalence data in studies contributing to meta-analysis. Note: Urban and rural values do not sum to the overall total because some surveys did not provide disaggregated results, even though they reported urban–rural prevalence. In addition, some surveys included other regional categories (e.g., pastoralist populations), which also contribute to the discrepancy. Percentages are calculated with respect to the reported urban–rural values.
(CSV)

**S2 Fig. Urban and rural populations in 26 study countries between 2000 and 2024.** Urban and rural population data obtained from United Nations Populations 2024 (https://population.un.org/wpp/).
(TIFF)

**S3 Fig. Urban and rural populations percentages in 26 study countries between 2000 and 2024.** Urban and rural population data obtained from United Nations Populations 2024 (https://population.un.org/wpp/).
(TIFF)

**S4 Fig. Estimated TB incidence in 26 study countries between 2000 and 2024.** Black circles are central estimates, reported by WHO; blue line and bands are estimated from a Bayesian multivariate regression model of incidence and case detection ratio data.
(TIFF)

**S5 Fig. Estimated TB case detection ratio in 26 study countries between 2000 and 2024.** Black circles are central estimates, reported by WHO; purple line and bands are estimated from a Bayesian multivariate regression model of incidence and case detection ratio data.
(TIFF)

**S6 Fig. Predicted urban-to-rural prevalence ratio in bacteriologically-confirmed TB in 26 study countries: 2000–2024.** Country-specific predictions of time trends in urban-to-rural bacteriologically-confirmed TB, estimated from a Bayesian meta-analysis model
(TIFF)

**S5 Table. Estimated numbers and percentage of prevalent TB in urban and rural areas in 26 study countries between 2000 and 2024.** Estimated from a Bayesian multivariate regression model, which jointly estimated TB incidence and case detection ratio, and combined with assumptions around duration of infectiousness, distribution of urban and rural populations, with modelled estimates of urban and rural TB prevalence by year.
(XLSX)

**S7 Fig. Estimated numbers of people with prevalent TB in urban and rural areas in 26 study countries between 2000 and 2024.** Estimated from a Bayesian multivariate regression model, which jointly estimated TB incidence and case

detection ratio, and combined with assumptions around duration of infectiousness, distribution of urban and rural populations, with modelled estimates of urban and rural TB prevalence by year.
(TIFF)

## Acknowledgments

The views expressed are those of the author(s) and not necessarily those of the NIHR or the Department of Health and Social Care.

The views expressed do not necessarily reflect the UK government's official policies.

## Author contributions

**Conceptualization:** Seyed Alireza Mortazavi, Nicole A. Swartwood, Nanki Singh, Katherine C. Horton, Nicholas A. Menzies, Peter MacPherson.

**Data curation:** Seyed Alireza Mortazavi, Nicole A. Swartwood, Nanki Singh, Melike Hazal Can, Hening Cui, Do Kyung Ryuk, Peter MacPherson.

**Formal analysis:** Seyed Alireza Mortazavi, Nicole A. Swartwood, Peter MacPherson.

**Funding acquisition:** Peter MacPherson.

**Investigation:** Seyed Alireza Mortazavi, Nicole A. Swartwood, Nanki Singh, Melike Hazal Can, Hening Cui, Do Kyung Ryuk, Katherine C. Horton, Nicholas A. Menzies, Peter MacPherson.

**Methodology:** Seyed Alireza Mortazavi, Nicole A. Swartwood, Katherine C. Horton, Nicholas A. Menzies, Peter MacPherson.

**Project administration:** Nicholas A. Menzies, Peter MacPherson.

**Resources:** Peter MacPherson.

**Software:** Seyed Alireza Mortazavi, Nicole A. Swartwood, Peter MacPherson.

**Supervision:** Katherine C. Horton, Nicholas A. Menzies, Peter MacPherson.

**Validation:** Nicholas A. Menzies, Peter MacPherson.

**Visualization:** Peter MacPherson.

**Writing – original draft:** Seyed Alireza Mortazavi, Peter MacPherson.

**Writing – review & editing:** Seyed Alireza Mortazavi, Nicole A. Swartwood, Nanki Singh, Melike Hazal Can, Hening Cui, Do Kyung Ryuk, Katherine C. Horton, Nicholas A. Menzies, Peter MacPherson.

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
