## [Editor Report · Decision Letter 0]

24 Sep 2025

Dear Dr MacPherson,

Thank you for submitting your manuscript entitled "Urban and rural prevalence of tuberculosis in low- and middle-income countries: a systematic review and meta-analysis" for consideration by PLOS Medicine.

Your manuscript has now been evaluated by the PLOS Medicine editorial staff as well as by an academic editor with relevant expertise and I am writing to let you know that we would like to send your submission out for external peer review.

For clinical studies, please upload a copy of your trial study protocol as a supporting information file. The study protocol should be the version submitted for approval to the institutional review board or ethics committee, should include any amendments to the study protocol, as well as the date of their approval by the institutional review or ethics committee. Please also detail any deviations from the study protocol in the Methods section of your manuscript. The editors will consider the protocol and study conduct prior to a final decision for external review.

Please re-submit your manuscript within two working days, i.e. by Sep 26 2025 11:59PM.

Kind regards,

Andreia Cunha, PhD

Senior Editor

PLOS Medicine

---

## [Decision Letter · Decision Letter 1]

1 Dec 2025

Dear Dr MacPherson,

My sincere apologies for the delay in getting back to you with a decision, which was due to challenges in securing all the necessary advice. Many thanks for submitting your manuscript "Urban and rural prevalence of tuberculosis in low- and middle-income countries: a systematic review and meta-analysis" (PMEDICINE-D-25-03336R1) to PLOS Medicine. The paper has been reviewed by subject experts and a statistician; their comments are included below and can also be accessed here: [LINK]

As you will see, the reviewers find your work of great interest, and they raise some important points that we must ask you to address in full. After discussing the paper with the editorial team and an academic editor with relevant expertise, I'm pleased to invite you to revise the paper in response to the reviewers' comments. We plan to send the revised paper to some or all of the original reviewers, and we cannot provide any guarantees at this stage regarding publication.

We ask that you submit your revision by Jan 05 2026 11:59PM. However, if this deadline is not feasible, please contact me by email, and we can discuss a suitable alternative.

Don't hesitate to contact me directly with any questions (acunha@plos.org).

Best regards,

Andreia

Andreia Cunha, PhD

Senior editor

PLOS Medicine

acunha@plos.org

Comments from the reviewers:

Reviewer #1:

PMEDICINE-D-25-03336R1

Mortazavi et al.

This is a very well-written manuscript on an important topic: urban-rural distribution in the prevalence of tuberculosis in LMIC. Based on Bayesian meta-analysis of prevalence survey (TBPS) data, the study provides relevant insights. In Africa, prevalence is shown to be higher in urban than in rural settings whereas in the Western pacific region prevalence tends to be higher in rural settings. The proportion of prevalent tuberculosis that is in urban settings has increased over the past two decades, which is only partially due to increasing urban populations. The tables and figures are clear.

I had a few comments that I hope will further strengthen the manuscript.

Major comments

1. The authors acknowledge the limitation that urban-rural distinction may mean different things in different regions and countries. An additional limitation in this regard may be that population denominators for tuberculosis prevalence estimates are generally based on census data and where those are not recent, on intercensal growth rates. These intercensal growth rates may not reflect recent changes, for example because due to rapid urbanization and urban-rural migration, the growth of urban populations is steeper than based in intercensal interpolation. I think this potential limitation should be acknowledged as well.

2. The prevalence estimates used in the meta-analyses were as much as possible based on adjustments made based on WHO-recommended analyses. A key adjustment here is inverse probability weighting for age-sex specific non-participation rates. As participation rates in TB prevalence surveys tend to be particularly low among young men who are often also the age-sex stratum with the highest TB prevalence, these adjustments have had large (upward) impact on the prevalence estimates from national TBPS for some countries, e.g. Nigeria and Philippines. However, the implicit underlying assumption that non-participation is random may not hold true, e.g. due to healthy worker bias (those who are healthy are at work and don't show up for the survey). As young men will be often be the ones that migrate to urban areas this may bias the proportion TB that is urban versus rural. I would therefore suggest that the investigators add a sensitivity analysis where the unadjusted estimates from the surveys are used rather than the adjusted estimates.

3. If I understand it correctly, the Statistical modelling of incidence and case detection was based on WHO estimates of country-specific TB incidence and case detection rates where the urban-rural distribution of TB prevalence was derived from the meta-analysis. The WHO estimates of TB incidence and case detection rate are however not independent of the prevalence survey results, for many countries they are directly based on them. Should that interdependence be considered a problem?

4. An associated potential limitation is that for a number of countries in the analyses, the WHO estimates of TB incidence are unchanged over the 19-year period (Fig S10). That seems odd, and may be a reflection of lack of data or other limitations on the part of WHO. I believe that the limitations of the WHO estimates should be mentioned as well.

Minor comments:

Line 230: "%" seems missing

Table 1, bottom row. Is this about symptom screening ONLY used for sputum eligibility (i.e. no chest X-ray screening)? In my view it should, because surveys that only did symptom screening will not detect asymptomatic TB and studies have suggested that asymptomatic TB is more common in urban settings.

Discussion, lines 327-330: "(…) as emphasised by the absence of association with urbanisation indicators in meta-regression". I may have missed it but where exactly is this in the Results? Table 1 only shows one such indicator (Percent of population in urban areas), I did not see other important indicators such as population density.

Reviewer #2: Dear Authors,

Thank you for the opportunity to review your manuscript, "Urban and rural prevalence of tuberculosis in low- and middle-income countries: a systematic review and meta-analysis." This is a timely and important study that addresses critical gaps in our understanding of TB epidemiology across diverse settings. The manuscript is well-structured and provides valuable insights into urban-rural differences in TB prevalence. I have provided detailed comments and suggestions below aimed at clarifying methods, improving consistency, and enhancing the overall clarity and impact of the manuscript.

Title:

The review aims to assess TB prevalence in low- and middle-income countries (LMICs). However, a few included studies are from upper-middle-income countries such as China and Mongolia. According to the World Bank classification, these countries are upper-middle-income, not low- or lower-middle-income. I strongly recommend that the authors clarify how country income status was defined. Alternatively, consider including studies from upper-middle-income countries and modifying the title to reflect broader economic settings, e.g., "Urban and rural prevalence of tuberculosis across diverse economic settings: a systematic review and meta-analysis."

Abstract:

In the Methods section, the authors mention including TB prevalence studies from LMICs between 1993 and 2024. However, in the Results subsection, the included surveys are reported as conducted between 2000 and 2019. Please clarify this discrepancy for readers.

Background:

The Background provides useful context but requires refinements to improve clarity and alignment with the study's aims:

1. Clearly define "LMICs" and indicate whether upper-middle-income countries (e.g., China) were included. If so, justify this early.

2. Explicitly highlight the knowledge gap, although national surveys report urban-rural differences, no global quantitative estimate exists.

3. Organize urban and rural determinants more clearly (urban factors together, rural factors together, then migration) to improve readability.

4. Provide a clearer rationale for quantifying urban-rural TB differences in terms of policy, resource allocation, and End TB targets.

Methods:

1. The manuscript mentions including studies published between 1 January 1993 and 1 January 2024. Please clarify why this period was selected and ensure consistency across sections.

2. Confirm that all included studies define adults as 15 years and above. If some studies use 18+, report these definitions.

3. Using WHO regions may be less informative than income-based stratification, which aligns more closely with your title and TB prevalence objectives.

4. I commend the authors for applying this method. It appropriately accounts for between-study variability, incorporates uncertainty, and allows inclusion of relevant covariates. This is a methodological strength.

Results:

1. Correct the total number of excluded studies in the PRISMA flow chart; currently, it shows 86, but it should be 155.

2. Correct minor typos and missing full stops (e.g., line 214).

3. The interpretation is not clear to readers. The Western Pacific region is reported with PR 0.64 (95% CrI: 0.45-0.89). Clarify that this indicates urban TB prevalence is 36% lower than rural prevalence for clearer interpretation.

4. Upper-middle-income countries are included despite the title claiming LMICs. Either remove these countries or include other upper-middle-income countries and modify the title accordingly. I recommend reanalysing subgroup analyses by World Bank income classification, which aligns with TB prevalence objectives. Use WHO regions only if focusing on geographic or regional patterns.

Discussion (General Comments):

The discussion is comprehensive, clearly linking urbanization, demographic changes, and TB prevalence. Regional heterogeneity and social determinants are well-addressed, and policy implications are highlighted. Minor improvements could enhance clarity: group similar points to reduce repetition and explicitly connect demographic trends to meta-analysis findings. Limitations are acknowledged, but a brief discussion of how they might affect urban-rural comparisons would strengthen interpretation. Overall, the discussion effectively conveys the public health relevance of the findings.

Reviewer #3: See attachment

Michael Dewey

---

* Please upload any figures associated with your paper as individual TIF or EPS files with 300dpi resolution at resubmission; please read our figure guidelines for more information on our requirements: http://journals.plos.org/plosmedicine/s/figures. While revising your submission, we strongly recommend that you use PLOS's NAAS tool (https://ngplosjournals.pagemajik.ai/artanalysis) to test your figure files. NAAS can convert your figure files to the TIFF file type and meet basic requirements (such as print size, resolution), or provide you with a report on issues that do not meet our requirements and that NAAS cannot fix.

After uploading your figures to PLOS's NAAS tool - https://ngplosjournals.pagemajik.ai/artanalysis, NAAS will process the files provided and display the results in the "Uploaded Files" section of the page as the processing is complete.

If the uploaded figures meet our requirements (or NAAS is able to fix the files to meet our requirements), the figure will be marked as "fixed" above. If NAAS is unable to fix the files, a red "failed" label will appear above.

When NAAS has confirmed that the figure files meet our requirements, please download the file via the download option, and include these NAAS processed figure files when submitting your revised manuscript.

FIGURES AND TABLES

SUPPLEMENTARY MATERIAL

REFERENCES

SYSTEMATIC REVIEWS & META-ANALYSES

* Please report your SR/MA according to the PRISMA guidelines provided at the EQUATOR site. http://www.equator-network.org/reporting-guidelines/prisma/. Please provide the completed PRISMA checklist as Supporting Information. When completing the checklist, please use section and paragraph numbers, rather than page numbers. Please add the following statement, or similar, to the Methods: "This study is reported as per the Preferred Reporting Items for Systematic Reviews and Meta-Analyses (PRISMA) guideline (S1 Checklist)."

* Abstract: Please report your abstract according to PRISMA for abstracts (https://doi.org/10.1371/journal.pmed.1001419) following the PLOS Medicine abstract structure (Background, Methods and Findings, Conclusions). Please ensure you provide dates of search, data sources, number of studies included, types of study designs included, eligibility criteria, and synthesis/appraisal methods.

* Please note that we expect searches to be updated to within 6 months of the time of submission.

---

## [Decision Letter · Decision Letter 2]

27 Feb 2026

Dear Dr. MacPherson,

Sincere apologies for the delay in getting back to you with a decision. Thank you very much for re-submitting your manuscript "Urban and rural prevalence of tuberculosis in low- and middle-income countries: a systematic review and meta-analysis" (PMEDICINE-D-25-03336R2) for review by PLOS Medicine.

I have discussed the paper with my colleagues and the academic editor and it was also seen again by the original reviewers. I am pleased to say that provided the remaining Reviewer 1, editorial and production issues are dealt with we are planning to accept the paper for publication in the journal.

[LINK]

We look forward to receiving the revised manuscript by Mar 06 2026 11:59PM.

Sincerely,

Andreia Cunha, PhD

Senior Editor

PLOS Medicine

plosmedicine.org

Requests from Editors:

GENERAL EDITORIAL REQUESTS

* At this stage, we ask that you include a short, non-technical Author Summary of your research to make findings accessible to a wide audience that includes both scientists and non-scientists. The Author Summary should immediately follow the Abstract in your revised manuscript. This text is subject to editorial change and should be distinct from the scientific abstract. Ideally each sub-heading should contain 2-3 single sentence, concise bullet points containing the most salient points from your study. In the final bullet point of ‘What Do These Findings Mean?’ Please include the main limitations of the study in non-technical language.

Please see our author guidelines for more information: https://journals.plos.org/plosmedicine/s/revising-your-manuscript#loc-author-summary."

* Please confirm that your title complies with PLOS Medicine's style. Your title must be nondeclarative and not a question. It should begin with main concept if possible. "Effect of" should be used only if causality can be inferred, i.e., for an RCT. Please place the study design ("A randomized controlled trial," "A retrospective study," "A modelling study," etc.) in the subtitle (ie, after a colon).

* Please confirm that your abstract complies with our requirements, including format (three sections: Background, Methods and Findings, and Conclusions) and providing all the information relevant to this study type https://journals.plos.org/plosmedicine/s/submission-guidelines#loc-abstract

* Please ensure that the Introduction ends with a clear description of the study question or hypothesis.

* Please ensure that all abbreviations are defined at first use throughout the text.

* Please confirm that all numbers presented in the abstract are present and identical to numbers presented in the main manuscript text.

GENERAL

* Please review your text for claims of novelty or primacy (e.g. 'for the first time') and remove this language. In addition, please check that any use of statistical terms (such as trend or significant) are supported by the data, and if not please remove them.

* Statistical reporting: Please revise throughout the manuscript, including tables and figures.

- Please report statistical information as follows to improve clarity for the reader "(95% CI [13,28]; p</=)".

- Please separate upper and lower bounds with commas instead of hyphens as the latter can be confused with reporting of negative values.

- Please repeat statistical definitions (HR, CI etc.) for each set of parentheses.

* In the abstract, please include the important dependent variables that are adjusted for in the analyses.

FUNDING STATEMENT

* The funding statement should include: specific grant numbers, initials of authors who received each award, URLs to sponsors’ websites. Also, please state whether any sponsors or funders (other than the named authors) played any role in study design, data collection and analysis, the decision to publish, or preparation of the manuscript. If they had no role in the research, include this sentence: “The funders had no role in study design, data collection and analysis, decision to publish, or preparation of the manuscript.”

COMPETING INTERESTS STATEMENT

* All authors must declare their relevant competing interests per the PLOS policy, which can be seen here: https://journals.plos.org/plosmedicine/s/competing-interests For authors with ties to industry, please indicate whether any of the interests has a financial stake in the results of the current study.

* Please add this statement to the manuscript's Competing Interests: "[Initials] is an Academic Editor on PLOS Medicine's editorial board."

FIGURES

* Please provide titles and legends for all figures and tables (including those in Supporting Information files). Please define all acronyms used in each figure or table in its corresponding legend.

* Please provide the unadjusted comparisons as well as the adjusted comparisons in all relevant Tables

* Please specify the variables controlled for in all relevant Tables

* Please ensure that where relevant figures include 95% CIs.

SRAs / MAs

* Please report your SR/MA according to the PRISMA guidelines provided at the EQUATOR site.

http://www.equator-network.org/reporting-guidelines/prisma/

Please provide the completed PRISMA checklist as Supporting Information.

* Please add the following statement, or similar, to the Methods: ""This study is reported as per the Preferred Reporting Items for Systematic Reviews and Meta-Analyses (PRISMA) guideline (S1 Checklist)."""

* Please provide the beginning and end dates of your search.

* Please report your abstract according to PRISMA for abstracts, following the PLOS Medicine abstract structure (Background, Methods and Findings, Conclusions) http://www.plosmedicine.org/article/info:doi/10.1371/journal.pmed.1001419 .

Comments from Reviewers:

Reviewer #1: I thank the authors for their responses and their revisions. They have addressed all my comments well, with one exception.

Comment #1.2: Perhaps my comment was not clear, but both the adjusted and the unadjusted estimates potentially suffer from bias. The adjusted estimates indeed corrected for sampling bias through multiple imputation and inverse probability weighting. Both methods assume that not participating in the survey is independent of the probability of prevalent TB. My point is that this may not be true, especially not for young adults and males (healthy worker bias), and that the extent of this dependence may be different between urban and rural settings. A sensitivity analysis based on unadjusted prevalence estimates would tell us about the robustness of the findings to this assumption of independence. Of course it would not provide the "correct" result, but that's not what sensitivity analyses are meant for. If such a sensitivity analysis is considered too cumbersome, the authors should acknowledge this potential source of bias.

Reviewer #2: Congratulations for your great contribution to the existing literature. Thank you for addressing all of my concerns. I have reviewed the revised version carefully, and the manuscript has substantially improved, and suitable for publication. I have no further comment.

Reviewer #3: The authors have fully addressed all my points.

Michael Dewey

[LINK]

---

## [Editor Report · Decision Letter 3]

23 Mar 2026

Dear Dr MacPherson,

On behalf of my colleagues and the Academic Editor, Dr Amitabh Bipin Suthar, I am pleased to inform you that we have agreed to publish your manuscript "Urban and rural prevalence of tuberculosis in low- and middle-income countries: a systematic review and meta-analysis" (PMEDICINE-D-25-03336R3) in PLOS Medicine.

PRESS

Sincerely,

Andreia Cunha, PhD

Senior Editor

PLOS Medicine